# Laboratory evaluation of open source and commercial electrical conductivity sensor precision and accuracy: How do they compare?

**Stephanie G. Fulton** [1,2]*, **James C. Stegen** [1,3], **Matthew H. Kaufman** [1], **John Dowd** [4], **Aaron Thompson** [2]

1 Pacific Northwest National Laboratory, Richland, Washington, United States of America, 2 Department of Crop and Soil Sciences, University of Georgia, Athens, Georgia, United States of America, 3 School of the Environment, Washington State University, Pullman, Washington, United States of America, 4 Geology Department, University of Georgia, Athens, Georgia, United States of America

These authors contributed equally to this work.
* stephanie.fulton@pnnl.gov

**Data Availability Statement:** The minimal data set and statistical analysis code underlying the findings in our study, along with the Arduino IDE OS software and instructions to operate the open

## Abstract

Variation in the electrical conductivity (EC) of water can reveal environmental disturbance and natural dynamics, including factors such as anthropogenic salinization. Broader application of open source (OS) EC sensors could provide an inexpensive method to measure water quality. While studies show that other water quality parameters can be robustly measured with sensors, a similar effort is needed to evaluate the performance of OS EC sensors. To address this need, we evaluated the accuracy (mean error, %) and precision (sample standard deviation) of OS EC sensors in the laboratory via comparison to EC calibration standards using three different OS and OS/commercial-hybrid (OS/C) EC sensors and data logger configurations and two commercial (C) EC sensors and data logger configurations. We also evaluated the effect of cable length (7.5 m and 30 m) and sensor calibration on OS sensor accuracy and precision. We found a significant difference between OS sensor mean accuracy (3.08%) and all other sensors combined (9.23%). Our study also found that EC sensor precision decreased across all sensor configurations with increasing calibration standard EC. There was also a significant difference between OS sensor mean precision (2.85 µS/cm) and the mean precision of all other sensors combined (9.12 µS/cm). Cable length did not affect OS sensor precision. Furthermore, our results suggest that future research should include evaluating how performance is impacted by combining OS sensors with commercial data loggers as this study found significantly decreased performance in OS/commercial-hybrid sensor configurations. To increase confidence in the reliability of OS sensor data, more studies such as ours are needed to further quantify OS sensor performance in terms of accuracy and precision across different settings and OS sensor and data collection platform configurations.

source (OS) Atlas electrical conductivity (EC) sensor and the user manuals to operate the YSI EcoSense® EC300A and Decagon CTD-10 commercial EC sensors, will be published at the U. S. Department of Energy's Environmental System Science Data Infrastructure for a Virtual Ecosystem (ESS-DIVE) repository (https://ess-dive.lbl.gov/about/). (DOI: 10.15485/1971118.) Fulton S G; Stegen J C; Kaufman M H; Dowd J; Thompson A (2023): Data and Scripts associated with: "Laboratory evaluation of open source and commercial electrical conductivity sensor precision and accuracy". River Corridor and Watershed Biogeochemistry SFA, ESS-DIVE repository. Dataset. doi:10.15485/1971118 accessed via https://data.ess-dive.lbl.gov/datasets/doi:10.15485/1971118 on 2023-04-24.

**Funding:** This research was supported in part by the U.S. Department of Energy (USDOE), Office of Science, Office of Biological and Environmental Research, Environmental System Science (ESS) Program (https://ess.science.energy.gov/). This contribution originates from the River Corridor Scientific Focus Area (SFA) project at Pacific Northwest National Laboratory (PNNL) funded under USDOE Grant No. 54737. SGF, JCS, and MHK received funding under USDOE Grant No 54737 to prepare this manuscript. This research was also supported in part by the Electric Power Research Institute (EPRI; https://www.epri.com/) under EPRI Contract Agreement No. 10003991 to The University of Georgia, which funded the purchase of equipment and supplies for this study. AAT is the Principal Investigator for EPRI Agreement No. 10003991 while SGF and JD worked with equipment purchased under this agreement. None of the funders listed herein had any role in study design, data collection and analysis, decision to publish, or preparation of the manuscript.

**Competing interests:** The authors have declared that no competing interests exist.

## Introduction

In situ sensors have become an increasingly important tool in the environmental sciences, particularly for monitoring water quality and quantity at a variety of scales across a diverse array of aquatic systems [1], including urban stormwater runoff collection systems [2, 3], wastewater treatment plants [2, 4], agriculture and irrigation systems [5–8], coastal monitoring [9], and snowpack and soil moisture measurements in mountain regions [10, 11]. Sensors and sensor networks provide high-frequency time series data that complement traditional water quality monitoring and assessment methods. Traditional methods are reliant upon time-consuming and expensive collection and laboratory analysis of physical water chemistry samples. In contrast, sensor-based water chemistry data can decrease costs associated with generation of time series data needed to capture hydrologic events and changes in water chemistry that cannot be captured with weekly or monthly grab samples [12–14].

Most water quality sensing is done via commercial sensors and data collection platforms, which are powerful tools but have limitations [15]. These systems can be expensive, and their cost can be a large impediment to collecting high spatial and temporal resolution water quality data needed by governmental and scientific organizations to effectively manage water resources across a range of scales. Furthermore, proprietary software and hardware required by commercial sensor vendors offer users a limited set of data collection and data management strategies, including data collection time intervals, data downloads, integration with a range of other sensors, and the incorporation of customized data post-processing and visualization tools [15].

Open source (OS) sensors and data collection platforms may make it possible to solve some of the limitations of commercial sensors and data loggers. Over the past two decades, the international user community has embraced the use of OS technologies to better manage scarce water resources due to the proliferation of inexpensive OS sensors and low-cost, low-power OS microcontrollers, such as those made by Arduino® or Arduino-compatible (i.e., microcontrollers designed with the same input/output (I/O) pin configuration) designers and manufacturers [15]. Open source manufacturers' disclosure of hardware and software details provides users the freedom to modify the instruments' configurations and functionalities. The online presence of active and highly engaged users around the world, often referred to as the maker community, contribute computer code and provide technical support for a wide range of OS environmental sensors and code libraries that simplify computer programming for first-time users.

Unlike many commercial data loggers and sensors, OS monitoring systems can interface with commercial and OS sensors in myriad configurations, ranging from strictly OS hardware- and software-based data loggers and sensors [16, 17] to OS-hybrid monitoring systems that are configured with a variety of commercial and OS data loggers and sensors [1, 18]. Due to their inherent flexibility, OS data loggers and sensors can collect high temporal frequency at broader spatial scales, for a wider variety of parameters, and at lower cost than expensive, proprietary commercial sensors and data loggers [19]. For example, Bórquez-López, Martínez-Córdova [20] developed a low-cost OS sensor network to measure, log, and display in real-time the temperature, pH, and dissolved oxygen (DO) concentrations for aquaculture systems in Mexico. As another example, to better manage Lake Victoria water resources in Tanzania and Uganda, Faustine, Mvuma [21] developed a OS sensor network WSN to collect and view in real-time, high temporal frequency lake water quality data. The rise of the maker movement has enabled governments, municipalities, researchers, and citizen scientists around the world that could not otherwise afford commercial sensors and data loggers to design, build, and deploy OS sensor networks to collect critical environmental monitoring data to keep their communities safe from environmental hazards [22].

Open source systems also have their limitations, and recent studies have only begun to evaluate these shortcomings in the context of water quality. Early performance evaluations of OS sensor networks focused on overall network performance by analyzing metrics such as packet delivery ratios and received signal strength indicators [10], sensor node placement strategies to maximize network stability and robustness and increase the spatial representativeness of data [11, 23], and the continuity and reliability of data collection activities [8, 24]. The number of publications on low-cost environmental sensing networks has been steadily increasing since 2004, which as Mao, Khamis [25] note, coincides with the public release of OS Arduino microcontrollers. More recent studies have quantified OS water quality sensor precision and accuracy compared to commercial instruments and have found no significant differences in the measured variables, including temperature, pH, dissolved oxygen (DO), and oxidation-reduction potential (ORP) [20, 26].

Compared to the amount of data available for other OS sensors for water quality monitoring, there have been few studies that have evaluated the performance of OS electrical conductivity (EC) sensors. Variation in electrical conductivity (EC), a key element of water quality, is important to reveal environmental disturbance and natural dynamics [27, 28]. Anthropogenic salinization is increasingly recognized worldwide as a cause of biological and chemical impairment in rivers and streams [29–32]. Groundwater-surface water mixing [33–35], mining impacts on surface water and groundwater quality [36–38], fossil fuel discharges [39], tidal influence [40, 41], and rainfall/runoff relationships [42–45] have all been studied using EC measurements. Broader application of OS EC sensors could be impactful to basic and applied science as a quick, easy, and inexpensive method to measure water quality. While studies have demonstrated that other OS water quality sensors can deliver water quality measurements with a precision and accuracy comparable to more expensive "scientific-grade" commercial sensors [20, 26, 46–50], a similar effort is needed to evaluate the performance of OS EC sensors.

As part of a broader evaluation of OS sensors for hydrological and water quality monitoring at a remote surface coal mine in Kentucky, USA [51], this study evaluated the accuracy and precision of OS EC sensors in the laboratory by comparing EC data collected using five OS and commercial sensor and data logger configurations, including an OS data logger and EC sensor, two commercial data loggers and EC sensors, and one OS data logger/commercial sensor hybrid system. We also evaluated the effect of cable length (7.5 m and 30 m) and sensor calibration on OS sensor accuracy and precision. Studies like this are needed to guide the selection and use of OS sensors for water quality monitoring systems when funding is limited but the need for high quality data is critical. Our results demonstrated that OS EC sensors have as good or better precision and accuracy, relative to commercial sensors.

## Materials and methods

### Sensor configurations and calibration

We evaluated three different open source (OS) and OS/commercial-hybrid (OS/C) EC sensors and data logger configurations and two commercial (C) EC sensors and data logger configurations: 1) OS Atlas Scientific Environmental Robotics (Atlas) EC K 0.1 probe kit (EC-KIT-0.1; Atlas Scientific Environmental Robotics; Long Island City, NY, USA) connected with a 7.5-m cable to an OS Arduino-based data logger (Arduino Mega 2560 Rev3; Arduino; Somerville, MA, USA) (OS Atlas, 7.5-m), 2) OS Atlas EC K 0.1 probe kit connected with a 30-m cable to an OS Arduino data logger (OS Atlas, 30-m), 3) commercial Decagon multiprobe (Decagon CTD-10 conductivity/temperature/depth sensor and 10 m cable with 3-wire pigtail adapter; Meter Group, Inc.; Pullman, WA, USA) configured with an OS Arduino data logger (OS/C

**Table 1. Open source and commercial sensor and data logger configurations and 2–pt calibration standards.**

| Sensor Configuration ID | Sensor Configuration | Data Logger | Sensor | Cable Length | 2-Point Calibration |
|---|---|---|---|---|---|
| Atlas, 7.5-m cable (narrow calibration) | OS | Arduino Mega 2560 | Atlas EC probe kit | 7.5 | 84 and 1413 |
| Atlas, 30-m cable (narrow calibration) | OS | Arduino Mega 2560 | Atlas EC probe kit | 30 | 84 and 1413 |
| Atlas, 7.5-m cable (wide calibration) | OS | Arduino Mega 2560 | Atlas EC probe kit | 7.5 | 84 and 2060 |
| Atlas, 30-m cable (wide calibration) | OS | Arduino Mega 2560 | Atlas EC probe kit | 30 | 84 and 2060 |
| OS/C Decagon (no calibration) | OS/C | Arduino Mega 2560 | Decagon CTD-10 multiprobe | 10 | NA |
| YSI (wide calibration) | C | YSI EcoSense® EC300A | EcoSense® EC300A | 1 | 84 and 2060 |
| Decagon (no calibration) | C | Decagon EM50R* | Decagon CTD-10 multiprobe | 10 | NA |

m, meters; OS, open source; C, commercial; OS/C, open source/commercial hybrid; EC, electrical conductivity (μS/cm); NA, not applicable. Each sensor configuration ID includes a) the sensor manufacturer, b) sensor cable length (in m), and c) the 2–point calibration width (narrow or wide); calibration standards units in μS/cm. A sensor configuration consists of an EC sensor (OS or C), a data logger (OS or C), and a cable (7.5 m or 30 m) connecting the sensor to a data logger. An OS sensor configuration means that the data logger and sensor were developed using publicly available OS hardware and software. A C sensor configuration means that the data logger and sensor were configured with commercial—and often proprietary—hardware and software. An OS/C sensor configuration means that the data logger was developed using publicly available OS hardware and software and the sensor was configured with commercial hardware and software. A "narrow calibration" means that the sensor was calibrated with a 2–point calibration using 84 μS/cm and 1413 μS/cm EC calibration standards. A "wide calibration" means that the sensor was calibrated with a 2–point calibration using 84 μS/cm and 2060 μS/cm EC calibration standards. EC measurements were temperature corrected to 25˚ C and recorded as specific conductance (SC, units in μS/cm).

Decagon), 4) commercial Decagon EC multiprobe configured with a commercial Decagon data logger (Decagon EM50R; Meter Group, Inc.; Pullman, WA, USA) (C Decagon), and 5) commercial YSI handheld EC meter and probe with 1-m cable (EcoSense® EC300A handheld sensor with 1-m cable; YSI Incorporated; Yellow Springs, OH, USA) (C or commercial YSI) (Table 1). We included the OS/C Decagon in our performance evaluation to validate that our implementation of the modified SDI-12 communication protocol used to collect data via the OS/C Decagon EC sensor was successful thereby minimizing the risk of potential data losses during a remote sensor deployment. We evaluated the effect of cable length on OS Atlas EC sensor performance to assess whether cable lengths employed in the field (7.5 m and 30 m) influenced sensor measurements.

The OS Atlas EC probe kit includes an EC probe with a 0.1 cell constant (K) and 1-m cable with male BNC connector; EZO™ Conductivity Circuit (model EZO-EC Version 3.7; Atlas Scientific Environmental Robotics; Long Island City, NY, USA), Electrically Isolated EZO Carrier Board ("old style" model L-ISCCB; Atlas Scientific Environmental Robotics; Long Island City, NY, USA); and two EC calibration standards (84 μS/cm and 1,413 μS/cm). The K 0.1 probe has two graphite conductors with a measurement range of 0.07 μS/cm to 50,000 μS/cm. According to manufacturer specifications, the voltage response is linear between ~5 μS/cm and 5,000 μS/cm with an accuracy of ±2% [52].

The EZO carrier board was pinned to the breadboard and sat between the Atlas EC sensor and the Arduino data logger (Fig 1). The carrier board comes equipped a female BNC adapter on one end to connect to the probe cable or a 7.5-m or 30-m probe extension cable (BNC male to BNC female Extension Cables (BNC-7.5 and BNC-30; Atlas Scientific Environmental Robotics; Long Island City, NY, USA) and six input/output (I/O) pins on the other end to connect the 6-pin EZO circuit. The EZO circuit is a small computer system designed to communicate with and transfer data from Atlas sensors in conjunction with an external microcontroller using either serial or I2C communication protocols. The probe cable's BNC connector was connected to one end of an extension cable while the other end was connected to the carrier board's BNC adapter. The carrier board's pins were then connected to the Arduino with jumper wires (Fig 1).

The OS/C Decagon sensor was configured with a 10-m cable and 3-wire pigtail adapter for use with the Arduino data logger using the SDI-12 communication protocol. The pigtail adapter was connected to jumper wires with snap-action 3-wire connectors (Snap-action 3-Wire Block Connector (12–28 AWG); Adafruit Industries; New York, NY, USA), which were then connected to the Arduino via the breadboard (Fig 1).

The commercial sensor configurations were more straightforward than the OS sensor configurations: 1) the YSI EC probe was connected to the YSI handheld meter by a 1-m cable with connector assembly and 2) the Decagon multisensor was connected to the RM50 data logger by a 10-m cable with a stereo plug (Fig 1).

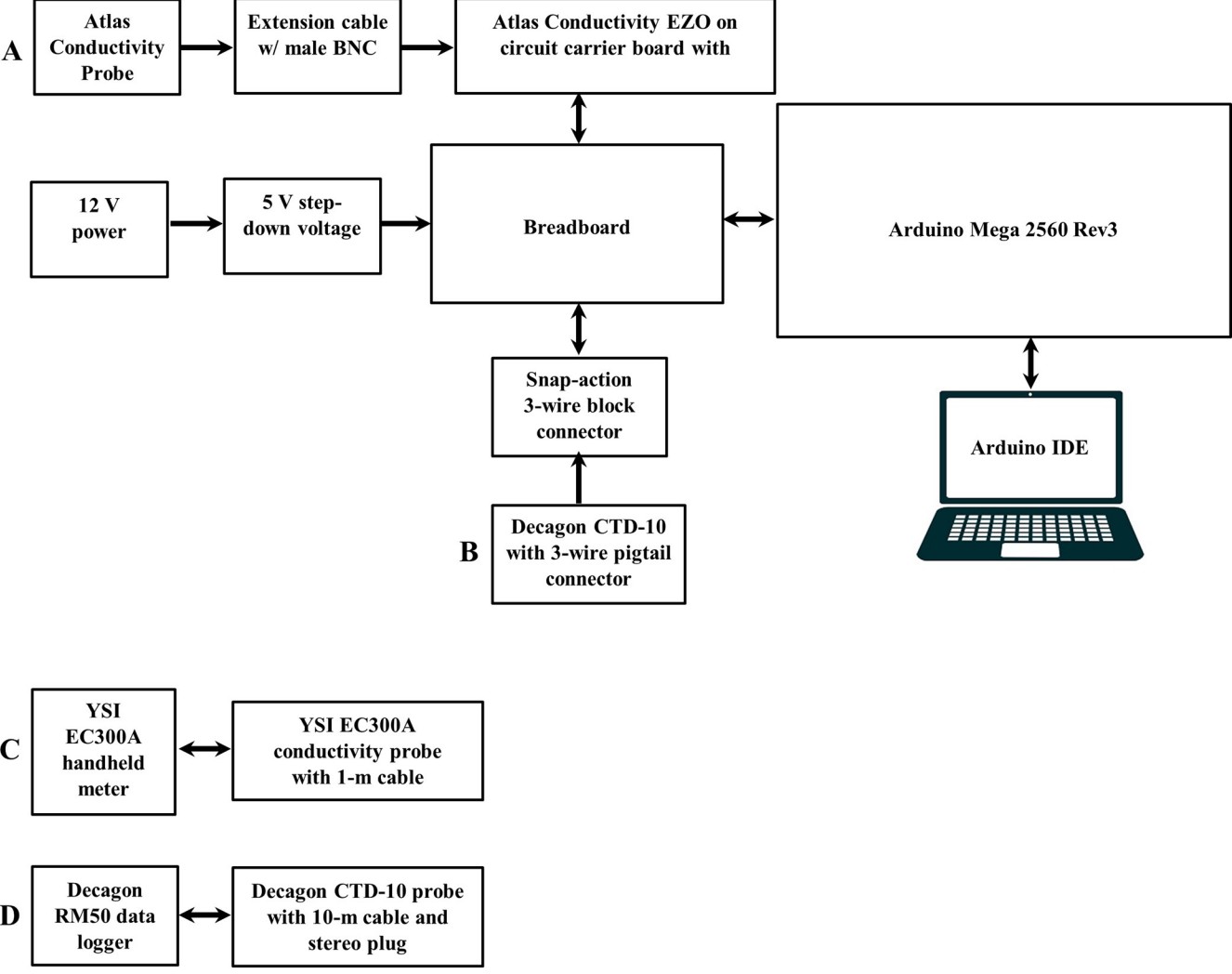

**Fig 1. Block diagram of sensor configurations evaluated for this study.** A) Open source (OS) Atlas Electrical Conductivity (EC) sensor configuration includes an Atlas EC probe connected to an Atlas Conductivity Circuit EZO and circuit carrier board using either a 7.5–m or 30–m extension cable and then connected to an OS Arduino Mega 2560 Rev3 data logger using jumper cables. B) OS/commercial (OS/C) hybrid Decagon EC sensor configuration includes a Decagon CTD–10 conductivity, temperature, and depth sensor and 10–m cable with a 3–wire pigtail connector connected to an OS Arduino Mega 2560 Rev3 data logger. C) Commercial YSI Instruments EC300A handheld meter and EC probe with a 1–m cable. Commercial Decagon EC sensor configuration includes a Decagon CTD–10 sensor with a 10–m cable with a stereo plug connected to a Decagon RM50 data logger. The OS Atlas and OS/C Decagon EC sensor configurations are powered directly by the Arduino, which is powered by an external 12 V power supply and a 5 V step–down voltage regulator connected to a breadboard using jumper cables.

Open source and OS/C EC sensor data collection and OS Atlas EC sensor calibrations were performed using an OS Arduino data logger programmed with the Arduino Integrated Development Environment (IDE) (Version 1.8.7; Arduino; Somerville, MA, USA). The Arduino, OS Atlas EC sensor, and OS/C Decagon multisensor were powered by a 12 V power supply (PS-38 AC/DC Power Supply; Electronix Express; Rahway, NJ, USA) by first stepping down the power to 5 V using a voltage regulator (5 V, 2.5 A Step-Down Voltage Regulator D24V22F5; Pololu Corporation; Las Vegas, NV, USA) pinned to the breadboard and then connecting the Arduino and sensors to the breadboard power and ground rails using jumper wires (Fig 1). The Arduino was programmed to calibrate and collect data with the OS Atlas EC sensor by modifying the sample code provided by Atlas to operate the EC sensor using an Arduino Mega [53] and the $I^2C$ communication protocol that works in conjunction with the Atlas EZO conductivity circuit [54] (see Data and code availability statement). Data was collected with the OS/C Decagon multiprobe by modifying an OS Arduino library for SDI-12 to work with the Arduino [55].

While most sensor manufacturers recommend using a 2-point calibration as the industry standard, some manufacturers recommended using a 3-point calibration when field conditions are expected to be in both low and high ranges [56] or if the calibration curve is not linear within the range of expected conditions. All OS Atlas and YSI sensor calibrations were performed immediately prior to the data collection according to manufacturer specifications using a 2-point calibration with EC calibration standards that bracketed the full range of expected conditions in the field at a remote mine site (i.e., ~200 to 2000 µS/cm) [51]. The YSI probe was calibrated according to the manufacturer-recommended 2-pt calibration using 84 µS/cm and 2060 µS/cm EC standards. The OS Atlas EC sensor was calibrated by first connecting a dry probe to the Arduino using either a 7.5-m or a 30-m sensor cable to the EZO circuit carrier board and then performing a 2-point calibration using one of two different paired calibration standards: 1) 84 µS/cm and 1413 µS/cm ("narrow calibration") or 2) 84 µS/cm and 2060 µS/cm ("wide calibration"), i.e., the same EC calibration standards recommended by the manufacturer for calibrating the YSI EcoSense EC300A. The Decagon CTD-10 sensor has an embedded chip that uses proprietary software that is inaccessible to users use to recalibrate the sensor. Like many other commercial conductivity sensors, the CTD-10 is factory calibrated and does not require further calibration.

## Data collection and processing

Three EC (µS/cm) measurements were collected following calibration over a 15-min period (one measurement every 5 min) using each sensor configuration. The Atlas, YSI, and Decagon EC sensors were placed together in a 500 mL beaker containing one of the EC calibration standards and repeated measurements were collected. Because the Atlas EC sensor does not have a temperature sensor, EC measurements were temperature compensated to 25˚ C during data collection using OS Decagon temperature data, which was collected immediately before each Atlas EC measurement, and recorded as SC (µS/cm). Electrical conductivity measurements collected with the commercial Decagon and YSI EC sensors are automatically temperature compensated to 25˚ C by the manufacturers' embedded proprietary software and recorded as SC (µS/cm). The flowchart shown in Fig 2 outlines the experimental design.

## Sensor accuracy and precision

We defined accuracy as the mean error (%) between the sensor measurement of the EC standard being measured (µS/cm) and the certified laboratory value of the calibration standard

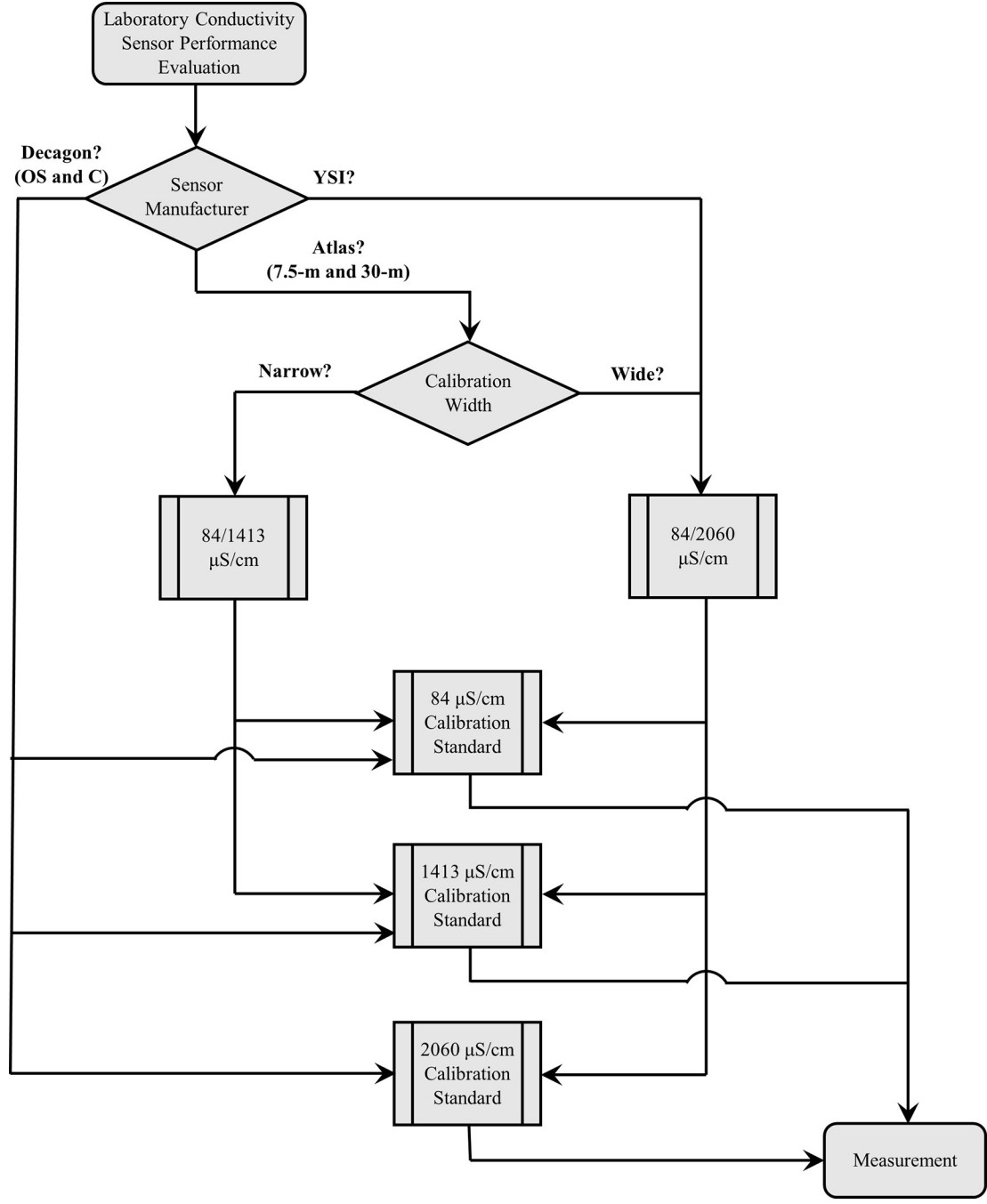

**Fig 2. Flowchart for evaluating electrical conductivity sensor accuracy and precision.**

(μS/cm) [20]. The mean error for each set of samples was defined as:

$$mean\ error = \frac{\sum_i^n \frac{SpC_S - SpC_{CS}}{C_{CS}} x\ 100}{n},$$  (1)

where $i$ = sample number, $n$ = number of samples, $SpC_S$ = sensor SC measurement (μS/cm),

and $SpC_{CS}$ = laboratory certified calibration standard (μS/cm). Sensor precision is a measure of repeatability and was defined as the sample standard deviation, $s$ [20].

Due to the small sample size and the presence of outliers, we performed Mann Whitney U Tests (`wilcox.test`, $p$ = 0.05) using R Studio [57] to determine whether there were statistically significant differences between the Atlas OS EC sensor mean accuracy and mean precision ($\bar{s}$, uS/cm) and the mean accuracy and mean precision of all other sensors combined (OS/C and C).

## Results and discussion

### Sensor accuracy

With few exceptions, all EC sensors evaluated had high accuracy (low mean error) with mean errors generally less than 10% (Fig 3 and S1 Table). The OS Atlas EC sensor had the highest accuracy with mean errors ranging from 0.4% to 11.9%. There was a statistically significant difference ($W$ = 12, $p$ = 0.021) between Atlas EC sensor mean accuracy (mean error = 3.08%) and the mean accuracy of all other sensors combined (commercial YSI, commercial Decagon, and OS/C Decagon; mean error = 9.23%). Atlas EC sensor mean accuracy was closely followed by the commercial YSI sensor with mean errors ranging from 1.3% to 7.2% (Fig 3). The commercial and OS/C Decagon sensor configurations had the lowest accuracy, with commercial Decagon mean errors ranging from 5.2% to 13.7% and OS/C Decagon sensor mean errors ranging from 6.9% to 28.2% (Fig 3).

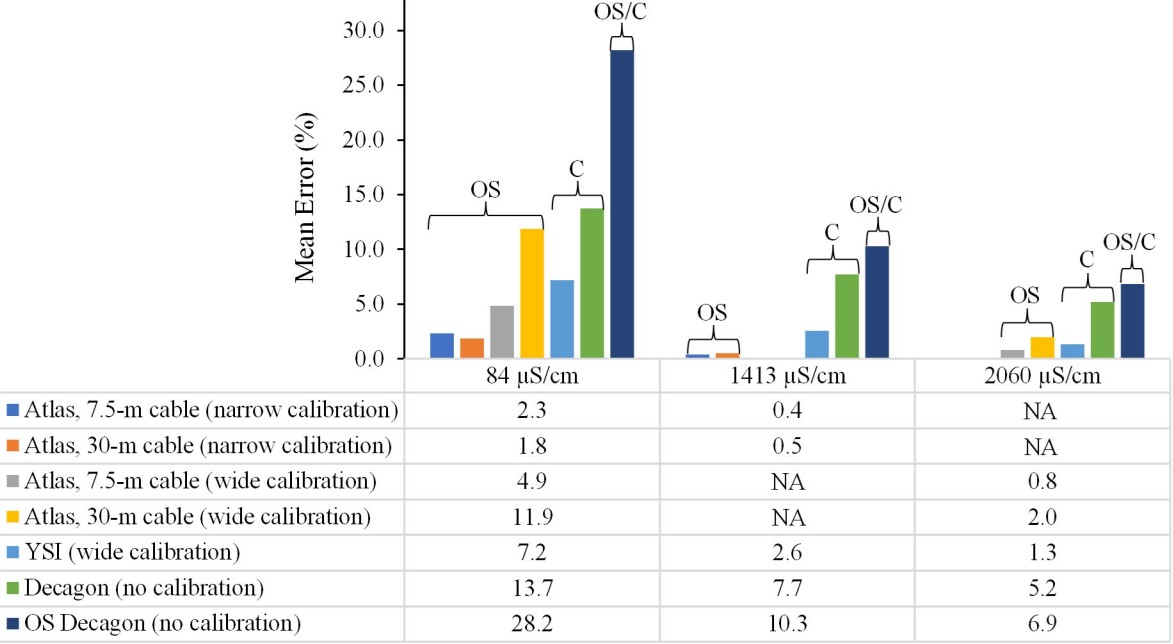

| | 84 μS/cm | 1413 μS/cm | 2060 μS/cm |
|---|---|---|---|
| ■ Atlas, 7.5-m cable (narrow calibration) | 2.3 | 0.4 | NA |
| ■ Atlas, 30-m cable (narrow calibration) | 1.8 | 0.5 | NA |
| ■ Atlas, 7.5-m cable (wide calibration) | 4.9 | NA | 0.8 |
| ■ Atlas, 30-m cable (wide calibration) | 11.9 | NA | 2.0 |
| ■ YSI (wide calibration) | 7.2 | 2.6 | 1.3 |
| ■ Decagon (no calibration) | 13.7 | 7.7 | 5.2 |
| ■ OS Decagon (no calibration) | 28.2 | 10.3 | 6.9 |

**Fig 3. The effect of sensor cable length and sensor calibration on electrical conductivity sensor accuracy.** OS, open source; C, commercial; OS/C, open source/commercial hybrid; m, meters; NA, not applicable (i.e., not measured). OS electrical conductivity (EC) sensor configurations include 1) Arduino data logger connected to an Atlas EC sensor with a 7.5–m cable and calibrated with a "narrow" 2–point calibration (i.e., 84 and 1413 μS/cm); 2) Arduino data logger connected to an Atlas EC sensor with a 30–m cable and calibrated with a "narrow" 2–point calibration (i.e., 84 and 1413 μS/cm); 3) Arduino data logger connected to an Atlas EC sensor with a 7.5–m cable and calibrated with a "wide" 2–point calibration (i.e., 84 and 2060 μS/cm); and 4) Arduino data logger connected to an Atlas EC sensor with a 30–m cable and calibrated with a "wide" 2–point calibration (i.e., 84 and 2060 μS/cm). EC measurements were temperature corrected to 25˚ C and recorded as specific conductance (SC, units in μS/cm).

Other OS Atlas non-EC water quality sensors—including temperature, pH, DO, and ORP sensors—have often been integrated into OS water quality monitoring systems [18, 58, 59]. Relatively consistent results across studies have shown that non-EC Atlas sensors also perform as well as or better than commercial options. For example, Bórquez-López, Martínez-Córdova [20] conducted a laboratory evaluation of a prototype OS water quality monitoring system developed for precision aquaculture by comparing the accuracy of Atlas OS pH and DO sensors and commercial pH and DO sensors. The authors found no significant differences ($p \geq 0.05$) in the variables measured by the OS and commercial sensors. In another quantitative evaluation of OS sensor performance compared to commercial sensors, Méndez-Barroso, Rivas-Márquez [60] designed a low-cost water quality monitoring system using Atlas OS pH, DO, and ORP sensors connected to an Arduino Nano-based data logger and evaluated Atlas sensor accuracy. Unlike our study, the authors calibrated the Atlas EC sensor over a much wider range of electrical conductivities by preparing eight non-standard sodium chloride (NaCl) calibration solutions ranging in electrical conductivities from 7690 to 61,548 μS/cm. Despite methodological and technological differences in their study, Méndez-Barroso, Rivas-Márquez [60] found results similar to ours whereby OS Atlas EC sensor accuracy was slightly better in the laboratory than the commercial EC sensor integrated in a YSI multiparameter probe (YSI-556; YSI Inc.; Yellow Springs, OH, USA). The authors also found that OS Atlas pH and DO sensors had similar accuracy as commercial pH and DO sensors integrated with the YSI multiparameter probe [60]. Our results similarly show that the accuracy of OS Atlas sensors is better or as good as commercial options. We caution, however, that pairing OS data loggers with commercial sensors (e.g., the Decagon CTD-10) may lead to decreased accuracy; we suggest avoiding such configurations if possible.

## Sensor precision

As a general pattern, EC sensor precision decreased (increasing sample standard deviation, $s$, μS/cm) across all sensor configurations with increasing calibration standard ionic strength (Table 2, Figs 4–6). The OS Atlas EC sensor was the most precise of the five sensor configurations evaluated, regardless of calibration width and calibration standard ionic strength (Figs 4–6). There was a statistically significant difference ($W = 15$, $p = 0.048$) between Atlas EC sensor mean precision (2.85 μS/cm) and the mean precision of all other sensors combined (9.12 μS/cm). OS Atlas EC sensor mean precision ranged from 0.2 μS/cm for the lowest ionic strength EC standard (30-m cable, wide calibration, 84 μS/cm) to 8.4 μS/cm for the highest ionic strength EC standard (30-m cable, wide calibration, 2060 μS/cm) (Table 2, Figs 4–6). OS Atlas EC sensor mean precision was greatest when measuring low ionic strength standards with a 30-m sensor cable and wide calibration but the magnitude and the direction of the change in precision did not respond consistently with changing calibration width (Table 2, Figs 4–6). Cable length did not affect OS Atlas EC sensor precision (Table 2, Figs 4–6).

Precision for C and OS/C configurations was similar in magnitude to OS sensor precision at the lowest ionic strength calibration standard (84 μS/cm), but the magnitude and direction (smaller $s$ vs. larger $s$) of the change in precision varied as ionic strength increased and by sensor configuration (Table 2 and Fig 4). YSI sensor precision ranged from 2.5 μS/cm to 13 μS/cm for the lowest (84 μS/cm) and highest (2060 μS/cm) calibration standards, respectively (Table 2). Both Decagon sensor configurations (C and OS/C) had the lowest precision of the sensor configurations evaluated, but OS/C precision was greater than C precision by nearly an order of magnitude for the lowest ionic strength standard (0.6 μS/cm and 3.8 μS/cm for OS/C and C, respectively) and the highest ionic strength calibration standard (3.1 μS/cm and 26 μS/cm for OS/C and C, respectively) (Table 2).

**Table 2. Open source and commercial electrical conductivity sensor precision.**

| Sensor Configuration[1] | Reference Standard | | |
|---|---|---|---|
| | 84 µS/cm | 1413 µS/cm | 2060 µS/cm |
| OS Atlas, 7.5-m cable (narrow calibration) | 1.0 | 2.1 | NA |
| OS Atlas, 30-m cable (narrow calibration) | 2.3 | 1.7 | NA |
| OS Atlas, 7.5-m cable (wide calibration) | 0.7 | NA | 6.4 |
| OS Atlas, 30-m cable (wide calibration) | 0.2 | NA | 8.4 |
| Commercial YSI (wide calibration) | 2.5 | 3.8 | 13 |
| Commercial Decagon (no calibration) | 3.8 | 9.3 | 26 |
| OS/C Decagon (no calibration) | 0.6 | 20 | 3.1 |

m, meters; OS, open source; OS/C, open source/commercial hybrid.

[1]A "sensor configuration" consists of an EC sensor (OS or Commercial), a data logger (OS or C), and a cable connecting the sensor to a data logger (length in m; either 7.5 m or 30 m).

An OS sensor configuration means that the data logger and sensor were developed using publicly available OS hardware and software. A commercial sensor configuration means that the data logger and sensor were configured with commercial—and often proprietary—hardware and software. An OS/C sensor configuration means that the data logger was developed using publicly available and OS hardware and software and the sensor was configured with commercial hardware and software. A "narrow calibration" means that the sensor was calibrated with a 2–point calibration using 84 µS/cm and 1413 µS/cm EC calibration standards. A "wide calibration" means that the sensor was calibrated with a 2–point calibration using 84 µS/cm and 2060 µS/cm EC calibration standards. EC measurements were temperature corrected to 25˚ C and recorded as specific conductance (SC, units in µS/cm).

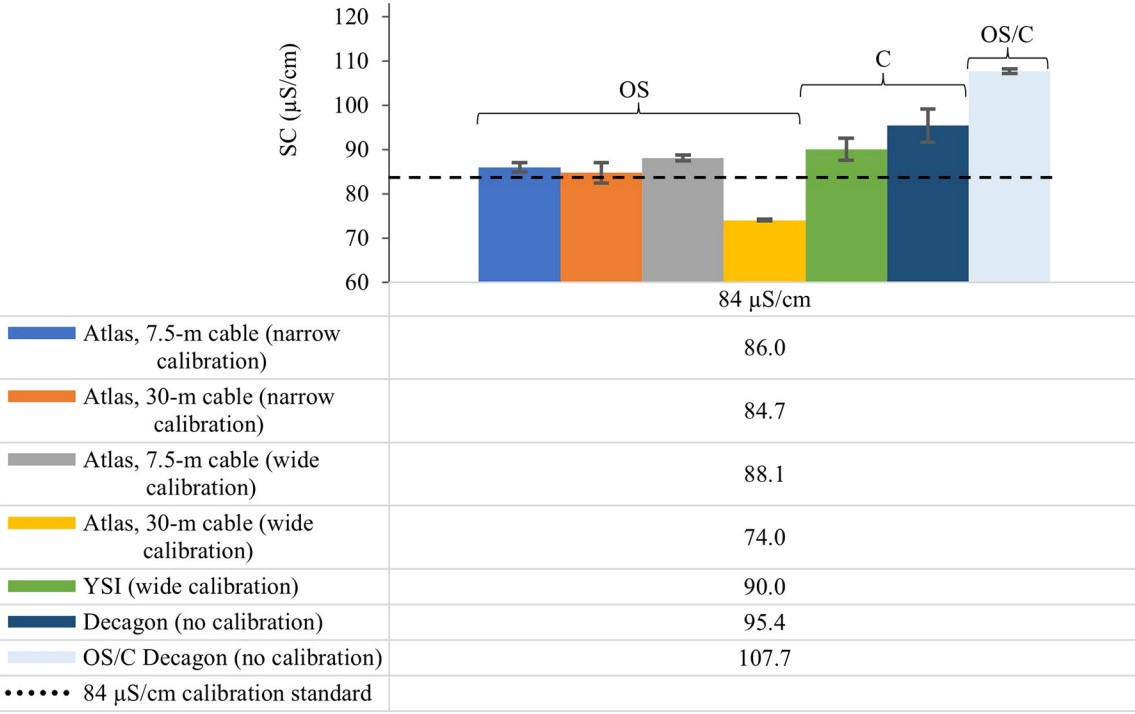

| | 84 µS/cm |
|---|---|
| Atlas, 7.5-m cable (narrow calibration) | 86.0 |
| Atlas, 30-m cable (narrow calibration) | 84.7 |
| Atlas, 7.5-m cable (wide calibration) | 88.1 |
| Atlas, 30-m cable (wide calibration) | 74.0 |
| YSI (wide calibration) | 90.0 |
| Decagon (no calibration) | 95.4 |
| OS/C Decagon (no calibration) | 107.7 |
| •••••• 84 µS/cm calibration standard | |

**Fig 4. The effect of sensor cable length and sensor calibration on electrical conductivity sensor precision when measuring the 84 µS/cm reference standard.** OS, open source; C, commercial; OS/C, open source/commercial hybrid; m, meters. OS electrical conductivity (EC) sensor configurations include 1) Arduino data logger connected to an Atlas EC sensor with a 7.5–m cable and calibrated with a "narrow" 2–point calibration (i.e., 84 and 1413 µS/cm); 2) Arduino data logger connected to an Atlas EC sensor with a 30–m cable and calibrated with a "narrow" 2–point calibration (i.e., 84 and 1413 µS/cm); 3) Arduino data logger connected to an Atlas EC sensor with a 7.5–m cable and calibrated with a "wide" 2–point calibration (i.e., 84 and 2060 µS/cm); and 4) Arduino data logger connected to an Atlas EC sensor with a 30–m cable and calibrated with a "wide" 2–point calibration (i.e., 84 and 2060 µS/cm). EC measurements were temperature corrected to 25˚ C and recorded as specific conductance (SC, units in µS/cm).

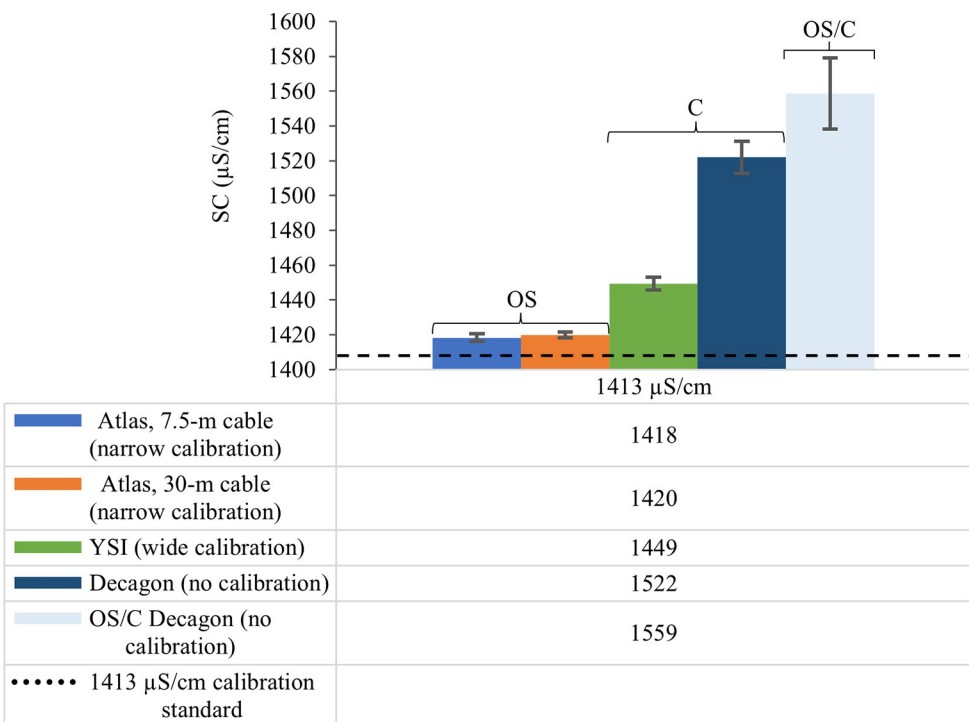

**Fig 5. The effect of sensor cable length and sensor calibration on electrical conductivity sensor precision when measuring the 1413 µS/cm reference standard.** OS, open source; C, commercial; OS/C, open source/commercial hybrid; m, meters. OS electrical conductivity (EC) sensor configurations include 1) Arduino data logger connected to an Atlas EC sensor with a 7.5–m cable and calibrated with a "narrow" 2–point calibration (i.e., 84 and 1413 µS/cm); 2) Arduino data logger connected to an Atlas EC sensor with a 30–m cable and calibrated with a "narrow" 2–point calibration (i.e., 84 and 1413 µS/cm); 3) Arduino data logger connected to an Atlas EC sensor with a 7.5–m cable and calibrated with a "wide" 2–point calibration (i.e., 84 and 2060 µS/cm); and 4) Arduino data logger connected to an Atlas EC sensor with a 30–m cable and calibrated with a "wide" 2–point calibration (i.e., 84 and 2060 µS/cm). EC measurements were temperature corrected to 25˚ C and recorded as specific conductance (SC, units in µS/cm).

Méndez-Barroso, Rivas-Márquez [60] conducted a field test of their low-cost coastal water quality monitoring multiparameter sonde by comparing OS sensor (Atlas pH, DO, and EC sensors) measurements to sensor measurements collected with a commercial multiparameter sonde (YSI EXO3, YSI Inc.; Yellow Springs, OH, USA) in Mobile Bay, Alabama, USA. Overall, the authors found that the OS multiparameter sonde had high precision (low root mean square error, RMSE) compared to the YSI measurements (as the "truth observation"). The low-cost sonde's OS temperature, salinity (OS Atlas EC voltage measurements converted to either parts per thousand [ppt] or practical salinity units [PSU] using the EZO EC circuit), and depth sensor measurements showed the highest precision of the parameters measured, while the OS EC, total dissolved solids (TDS, mg/L), and DO measurements had lower precision (higher RMSE) compared to the commercial sonde. The authors suggest that sediment deposition and biofouling due to differences in the location of the OS and commercial sensors could explain the lower precision in the OS sensor EC, TDS, and DO measurements [60].

## Conclusions

Open source water quality sensors and data loggers are rapidly becoming a user-friendly, low-cost, flexible, accurate, and dependable alternative to more expensive commercial data loggers and environmental sensors. Furthermore, the number and type of low-cost, off-the-shelf OS EC sensor options available today are far more numerous and less expensive than when we

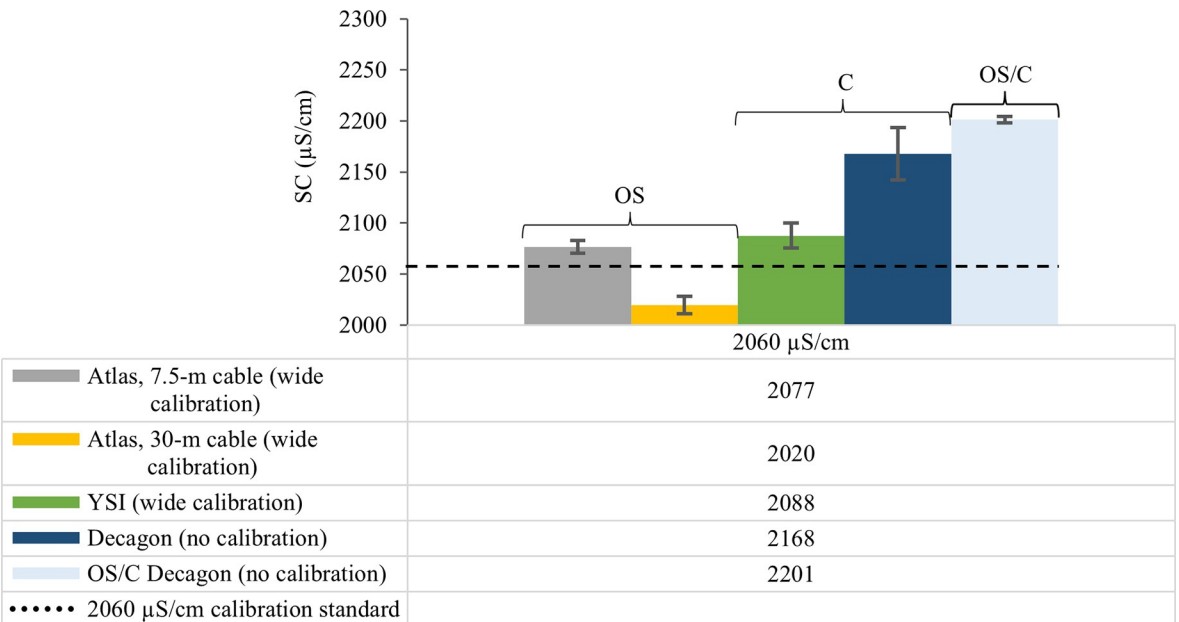

**Fig 6. The effect of sensor cable length and sensor calibration on electrical conductivity sensor precision when measuring the 2060 µS/cm reference standard.** OS, open source; C, commercial; OS/C, open source/commercial hybrid; m, meters. OS electrical conductivity (EC) sensor configurations include 1) Arduino data logger connected to an Atlas EC sensor with a 7.5–m cable and calibrated with a "narrow" 2–point calibration (i.e., 84 and 1413 µS/cm); 2) Arduino data logger connected to an Atlas EC sensor with a 30–m cable and calibrated with a "narrow" 2–point calibration (i.e., 84 and 1413 µS/cm); 3) Arduino data logger connected to an Atlas EC sensor with a 7.5–m cable and calibrated with a "wide" 2–point calibration (i.e., 84 and 2060 µS/cm); and 4) Arduino data logger connected to an Atlas EC sensor with a 30–m cable and calibrated with a "wide" 2–point calibration (i.e., 84 and 2060 µS/cm). EC measurements were temperature corrected to 25˚ C and recorded as specific conductance (SC, units in µS/cm).

began developing an OS wireless sensor network to monitor EC for a related study [51]. For example, both Seeed Studio (Grove EC Sensor Kit SE-110020292; Seeed Studio; Shenzhen, China) and DFRobot (Gravity: Analog Electrical Conductivity Sensor/Meter V2; DFRobot; Shanghai, China) make OS EC sensor kits that are similar to but less expensive than the OS Atlas EC sensor kit we used for this study. Open source sensors offer distinct advantages compared to commercial data loggers and sensors: 1) low cost means that more locations can be monitored for a given investment; 2) they offer increased flexibility and control over how data is collected and transmitted, customization over the data QA/QC process, and increased flexibility in terms of the application programming interface; and 3) large maker communities can provide a greater level of software and hardware technical support at no cost to the user compared to costly proprietary software and commercial manufacturer sensor maintenance contracts.

There are also disadvantages to using OS sensors for water quality monitoring inherent to the adoption of new technologies. In our experience, the initial amount of start-up time needed to teach users how to design, build, program, and test OS sensor-based water quality monitoring platforms can be substantially greater than the time needed to implement off-the-shelf commercial sensors. Furthermore, we have found that there is also a certain amount of skepticism to overcome regarding the use of OS sensors in the field due to the paucity of data on sensor performance, although this is changing rapidly. This resistance to OS sensors can be alleviated through a rigorous comparison and evaluation of OS sensor precision and accuracy in the laboratory prior to, during, and after field or lab deployments against the documented and accepted performance of commercial sensors.

To increase confidence in the reliability of OS sensor data, we need more studies such as this one to evaluate OS sensor performance in terms of accuracy and precision across different settings, OS sensor and data collection platform configurations, and over longer periods of deployment. While it is not necessary to account for potential sensor calibration drift during short-term laboratory sensor performance evaluations such as our study, for longer-term sensor deployment performance evaluations we recommend that users follow proper procedures for operating and maintaining sensors for continuous water quality monitoring, including regular calibration checks and the collection of quality-assurance data to correct for calibration drift and sensor biofouling. For example, USGS [61] recommends that measurements of specific conductance standards "...that are within ±5 μS/cm of the certified values for measurements ≤100 μS/cm or ±3 percent of the certified values for measurements >100 μS/cm are considered accurate and do not warrant recalibration."

Based on our study results, we suggest that future research also include evaluating how performance is impacted by combining OS sensors with commercial data loggers as this study found significantly decreased performance in OS/commercial-hybrid sensor configurations. Quantifying OS sensor performance and accuracy and making the data publicly available would reduce uncertainty surrounding the reliability and accuracy of OS sensor measurements and lead to broader acceptance and use of OS sensors. Enhancing access to OS sensors by citizen scientists and in resource poor research settings would increase our ability to monitor a more diverse array of aquatic resources in more locations at greater spatial and temporal frequencies than is currently possible.

## Supporting information

**S1 Table. Electrical conductivity sensor data and open source, commercial, and open source/commercial hybrid sensor and data logger configurations used to calibrate sensors and collect specific conductance data.** A "sensor configuration" consists of an EC sensor (OS or Commercial), a data logger (OS or C), and a cable connecting the sensor to a data logger (length in meters, m; either 7.5 m or 30 m). An OS sensor configuration means that the data logger and sensor were developed using publicly available, OS hardware and software. A commercial sensor configuration means that the data logger and sensor were configured with commercial—and often proprietary—hardware and software. An OS-C hybrid sensor configuration means that 1) the data logger was developed using publicly available, OS hardware and software and 2) the sensor was configured with commercial hardware and software. A "narrow calibration" means that the sensor was calibrated with a 2-point calibration using 84 μS/cm and 1413 μS/cm EC calibration standards. A "wide calibration" means that the sensor was calibrated with a 2-point calibration using 84 μS/cm and 2060 μS/cm EC calibration standards. The data was used in our statistical analysis of the overall accuracy and precision of the open source (OS) Atlas EC sensor configurations ("Atlas", column 2) against all non-Atlas EC sensor configurations ("Other", column 2). EC, electrical conductivity; OS, open source; C, commercial; OS/C, open source/commercial hybrid; m, meters; NA, not applicable (i.e., not measured). EC measurements were temperature corrected to 25˚ C and recorded as specific conductance (SC, units in μS/cm).
(CSV)

## Acknowledgments

Research funding and support was provided by the University of Georgia (UGA) Graduate School and the Crop and Soil Sciences Department, College of Agricultural and

Environmental Sciences. Funding for equipment and travel was also provided by the Electric Power Research Institute (EPRI), Palo Alto, California. The authors would also like to thank the faculty and staff of the UGA Geology Department, Franklin College of Arts and Sciences, for providing laboratory and office space throughout this study, particularly Michael Lewis for his many hours of advice and assistance troubleshooting open source hardware design and development. This research was also supported in part by the U.S. Department of Energy (DOE), Office of Science, Office of Biological and Environmental Research, Environmental System Science (ESS) Program (https://ess.science.energy.gov/). This contribution originates from the River Corridor Scientific Focus Area (SFA) project at Pacific Northwest National Laboratory (PNNL). PNNL is operated by Battelle Memorial Institute for the U.S. DOE under Contract No. DE-AC05-76RL01830.

## Author Contributions

**Conceptualization:** Stephanie G. Fulton.

**Data curation:** Stephanie G. Fulton.

**Formal analysis:** Stephanie G. Fulton.

**Funding acquisition:** Stephanie G. Fulton, Aaron Thompson.

**Investigation:** Stephanie G. Fulton.

**Methodology:** Stephanie G. Fulton, John Dowd.

**Project administration:** Stephanie G. Fulton.

**Resources:** Stephanie G. Fulton, Aaron Thompson.

**Software:** Stephanie G. Fulton, John Dowd.

**Supervision:** Stephanie G. Fulton, James C. Stegen, Matthew H. Kaufman, Aaron Thompson.

**Validation:** Stephanie G. Fulton.

**Visualization:** Stephanie G. Fulton.

**Writing – original draft:** Stephanie G. Fulton.

**Writing – review & editing:** Stephanie G. Fulton, James C. Stegen, Matthew H. Kaufman, Aaron Thompson.

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
