## [Decision Letter · Decision Letter 0]

20 Sep 2022

PONE-D-22-14296Laboratory evaluation of open source and commercial electrical conductivity sensor precision and accuracy: How do they compare?PLOS ONE

Dear Dr. Fulton,

Thank you for submitting your manuscript to PLOS ONE. After careful consideration, we feel that it has merit but does not fully meet PLOS ONE’s publication criteria as it currently stands. Therefore, we invite you to submit a revised version of the manuscript that addresses the points raised during the review process.

We look forward to receiving your revised manuscript.

Kind regards,

Talib Al-Ameri, Ph.D

Academic Editor

PLOS ONE

Journal Requirements:

Reviewers' comments:

Reviewer's Responses to Questions

**Comments to the Author**

1. Is the manuscript technically sound, and do the data support the conclusions?

Reviewer #1: Yes

Reviewer #2: Yes

2. Has the statistical analysis been performed appropriately and rigorously? 

Reviewer #1: Yes

Reviewer #2: Yes

3. Have the authors made all data underlying the findings in their manuscript fully available?

Reviewer #1: Yes

Reviewer #2: Yes

4. Is the manuscript presented in an intelligible fashion and written in standard English?

Reviewer #1: Yes

Reviewer #2: Yes

5. Review Comments to the Author

Reviewer #1: Dear authors,

As a user for the Atlas EC sensor I have read your manuscript with interest.

I have several comments that I feel could significantly improve your manuscript and make it more useful to the community.

1. Provide on a GitHub or other repository the details of how to operate and log the Atlas sensor (or provide links to other who have done it).

2. 2pt calibration, while they may be industry standard are problematic as between every two points passes a single line and hence the calibration process itself is not useful in providing likely uncertainties. It would be useful if you could discuss this issue.

3. It is not clear why you mention the DO and pH evaluations of the Atlas. How are they relevant here?

4. mg/l are not appropriate salinity units (l. 320).

5. The comparison is not strictly fair as you do not calibrate the Decagon sensor. Why not mimic the calibration procedure by forcing the Decagon to agree with the two points (e.g. using a linear transformation) and then evaluated how well it does?

6. A very important point you do not address is sensor drift. How long after a calibration can we expect it to remain the same? If the Atlas drift much faster than the Decagon that is something your reader will want to know. If you cannot evaluate it with your data to date, at least mention it in the discussion.

7. Please provide an explanation of how your evaluation is different from that of (50).

Minor comments:

l. 182: 'that' should be 'than'.

Fi.g 3. and 5: 'OS/C' is smooshed.

Dear authors: I am often wrong. If you feel my review missed the point feel free to contact me (emmanuel.boss@maine.edu) and I will be more than happy, if convinced, to change my review.

Reviewer #2: The contect of this paper is great and well presented.

I am wondering why the study only consider one type of OS EC sensor. I think the aim of this paper is also to evaluate the performance of different OS EC units, and I believe there are many available designs out there on the market with a cost ranging between $10 and $400.

I do not think data logger will become a source of error no matter you use Arduino or commercial products. As long as you make the logger support the standard data communicatio protocal (e.g., serial communication, SDI-12), not sure why the data logger might not correctly handle the transmission.

The things above are only for your to consider and address. On top of that, I only have two minor suggestions:

Line 116-119: citations needed

Line 249-265: what’s the purpose of this paragraph? You are talking about EC sensor, why need to mention the pH and DO sensor from the same company?

6. PLOS authors have the option to publish the peer review history of their article (what does this mean?). If published, this will include your full peer review and any attached files.

Reviewer #1: **Yes: **Emmanuel Boss

Reviewer #2: No

---

## [Author Response · Author response to Decision Letter 0]

20 Mar 2023

Reviewer 1: I have incorporated all of your suggestions into my revision. They were very helpful. Thank you for your review.

Reviewer 2: I have incorporated all of your suggestions into my revision. They were very helpful. Thank you for your review.

---

## [Editor Report · Decision Letter 1]

16 Apr 2023

Laboratory evaluation of open source and commercial electrical conductivity sensor precision and accuracy: How do they compare?

PONE-D-22-14296R1

Dear Dr. Fulton,

We’re pleased to inform you that your manuscript has been judged scientifically suitable for publication and will be formally accepted for publication once it meets all outstanding technical requirements.

Kind regards,

Talib Al-Ameri, Ph.D

Academic Editor

PLOS ONE

---

## [Editor Report · Acceptance letter]

27 Apr 2023

PONE-D-22-14296R1 

Laboratory evaluation of open source and commercial electrical conductivity sensor precision and accuracy: How do they compare? 

Dear Dr. Fulton:

I'm pleased to inform you that your manuscript has been deemed suitable for publication in PLOS ONE. Congratulations! Your manuscript is now with our production department. 

Kind regards, 

on behalf of

Dr. Talib Al-Ameri 

Academic Editor

PLOS ONE